# The Impact of Team Identity and Gender on Free-Riding Responses to Fear and Cooperation Sustainability

**Laura Gomez-Ruiz**  **and María J. Sánchez-Expósito** *

Department of Accounting and Finance, Pablo de Olavide University, 41013 Seville, Spain; lmgomrui@upo.es
* Correspondence: mjsanexp@upo.es

**Abstract:** This study explores the interaction effect of team identity and gender on free-riding responses to fear and cooperation sustainability in a social dilemma situation. Based on differences in inequity aversion, risk preferences, and reaction to competition between men and women, we predict that team identity reduces free-riding behaviors among men when they feel fear to be exploited by others teammates that free-ride, but that it does not affect women in this way. Consequently, we also predict that the effect of team identity on cooperation sustainability differs between the two genders. We conducted an experiment in which dominant incentives to free-ride were held constant over 30 periods and where agents had to make a decision between cooperation and free-riding in each period. After each decision, agents received teammates' contribution and earnings, which facilitates that agents identify whether their team members free-ride. Our findings show no effect for team identity on free-riding response to fear among women. However, team identity affects free-riding response to fear among men, which positively impacts cooperation sustainability.

**Keywords:** gender; team identity; informal control mechanisms; social dilemma; cooperation sustainability; fear

---

## 1. Introduction

This paper examines the relationship between gender and team identity in a social dilemma context. The social dilemma has proven to be a useful metaphor for analyzing cooperative behavior in situations of conflict between multiple interdependent actors who share a common resource [1,2]. Social dilemmas are characterized by a conflict between immediate self-interest and long-term collective interest [1,3,4]. Considering that "sustainability refers to longevity, continuity, and capability to be maintained" [5] (p. 393), cooperative behaviors are positively related to organizational sustainability (i.e., organizational performance or economic sustainability), while self-interested behaviors are harmful. In fact, self-interested behaviors (i.e., free-riding) have been identified as one of the most damage dysfunctional behaviors for economic sustainability. If agents free-ride, suboptimal outcomes will be achieved at the organizational level. If all agents behave in this manner, then the collaboration itself is destined to fail and organizational performance will not be sustained. Therefore, understanding which factors contribute to increasing and maintaining cooperation is critically important for organizational sustainability [5–7].

This study focuses on social dilemma contexts where team members periodically receive feedback about teammates' contributions and earnings. We try to replicate current organizational contexts where companies disclose information related to agents' performance, contributions, awards, and rewards or earnings [8]. The presence of feedback facilitates the emergence of free-riding behaviors. Free-riding can be the result of greed and fear inequalities. Greed is the temptation to free-ride when others

cooperate. Fear is the motivation to avoid being exploited by others that free-ride [9]. Information about teammates' contributions and earnings allows individuals to evaluate if they are suffering inequalities, that is, how fairly they are treated by teammates [10]. If individuals observe that those teammates are free-riding, they may experience fear. Consequently, they can decide to behave as free-riders [11–13]. Therefore, if individuals respond to free-riding when suffering fear, cooperation sustainability will be reduced not only directly by the first free-riders, but also indirectly by affecting cooperation of others on the team [14,15].

Researchers have analyzed how to overcome free-riding behaviors in social dilemma contexts. One solution to the problem is to change the incentive and eliminate the dilemma [16]. This is what organizations do when designing formal control systems such as incentives or penalties and when using in different ways formal control systems [17]. Another solution is to create groups that foster cooperative norms instead of free-riding behaviors. This is what informal control systems try to do. Informal controls are not consciously designed and include unwritten norms, loyalties, shared values, organizational culture, commitment, or team identity. Team identity has been highlighted as an informal control mechanism that affects agents' behavior [18,19]. Free-riding can be reduced if organizations make the team more meaningful for individuals (i.e., through team identity) [20–22]. Along this line, there is consensus about the negative effect of team identity on free-riding behaviors and the positive effect on cooperation when agents face social dilemmas [19,20]. Nevertheless, employees' features (e.g., age, gender, background) [23] and also employees' attitudes (e.g., readiness to change, commitment) [24] can explain differences in employees' behavior in organizational and team contexts. In the present paper, we focus on gender for two reasons. First, women presence is increasing in work contexts. Second, previous research suggests that social aspects do not influence men and women in the same way, and mixed results can be found when analyzing the relationship between team identity and gender [25–27]. Brown-Kruse and Hummels [25] found that men react more strongly to social identity than women in a multiperiod public goods provision game. In contrast, replicating Brown-Kruse and Hummels [25], Cadsby and Maynes [28] found no differences according to gender. Solow and Kirkwood [26] suggest "the effects of gender and group identity on behavior are more complicated, involving the nature of the social groups involved".

This paper seeks to contribute to this line of research by analyzing agents' behavior in a multiperiod prisoner's dilemma game, which is characterized by two features. First, we mixed men and women in teams, replicating current work settings where the presence of women in organizations and, therefore, in teamwork contexts cannot be ignored [29,30]. Therefore, we avoided the creation of all-male or all-female groups. Second, we provided information about agents' contributions and earnings after each decision, allowing for the detection of free-riding behaviors and, therefore, the appearance of fear. We are interested in free-riding behaviors because of fear considering that they are harmful for cooperation sustainability and considering the still current debate about how team identity affects fear in social dilemma context [9,31,32]. Previous studies suggest mixed and, sometimes, incongruent agent responses to fear, depending on their level of team identity. For instance, Simpson [9] suggests that team identity does not increase cooperative behaviors when agents suffer fear. However, another line of research suggests that when team identity is present, agents lose the sense of ingroup competition; therefore, they will decrease the level of free-riding behavior, even when feeling fear [20,31].

In this paper, we seek to contribute to the debate about how team identity influences agents' responses to fear in social dilemma contexts, suggesting that agents' responses will be different depending on gender. We follow research that shows that women and men react differently to inequity, risk, and competition [33,34]. While competition is important for men, women are more concerned about inequity and risky situations than men. Integrating social identity theory [20,35] with gender literature [26,36,37], the present study suggests that team identity decreases men's reactions to fear, but not women's. Accordingly, we suggest that the effect of team identity on free-riding and cooperation sustainability differs between the two genders.

To test our hypotheses, we conducted a 2 × 2 (identity × gender) (between subject) × 30 (periods) (within subject) experiment with female and male participants randomly assigned to three-person teams [6]. Our results show that team identity decreases the negative fear reaction among men; however, no effects were found in this regard for women. As a result, team identity only increases cooperation sustainability in men but not in women.

Results from this study enhance our understanding of how team identity, a tool that can be used as an informal control system, and gender, one specific feature of employees, interact when influencing agents' behavior in a social dilemma situation. First, we aim to contribute to the line of research that analyzes the effects of gender and social identity on agents' behavior and which suggests that this relationship is not straightforward [25,26,28,37]. Contrary to Cadsby and Maynes [28] but in line with Brown-Kruse and Hummels [25], our results show that men react more strongly to social identity than women. We contribute to this line of research by suggesting that as men worried more about competition and women more about risk and inequity, increased free-riding behaviors in response to fear differ between women and men. Second, we seek to contribute to informal control mechanisms research by understanding situations and contexts where social identity effects are not found. Despite the existing consensus about the positive effect of social identity on team members' cooperative behavior, less is known about whether gender differences matter [36,37]. Organizations may consider employees' features and attitudes to understand how employees react to organizational tools and changes [23,24]. The present paper highlights the importance of gender differences when designing control systems, due to the increased presence of women in work contexts and due to different gender responses to team identity.

The remainder of the paper is organized as follows. In the next section, we develop the hypotheses. We outline the design of the experiment in the third section. We describe the results in the fourth section, and finally we discuss the findings and limitations of the study.

## 2. Literature Review and Hypotheses Development

### 2.1. Literature Review

Free-riding is the case in which people may benefit from collectively provided public goods, but do not contribute their fair share [38]. At organizational level, an example is a group project in which the contributions of each member improve the quality of the project, but each one is tempted to allocate his/her time to other activities, while hoping the other group members will work in the group project [38]. If every group member decides to free-ride, then, the group project will fail. Therefore, for organizational performance and economic sustainability, free-riding behaviors are harmful [6,39].

The setting analyzed in this study is an iterated prisoner's dilemma situation in which team members receive information about teammates' contributions and earnings. If an agent realizes that teammates free-ride, the agent will experience a situation of fear. In this situation, free-riding behaviors can spread throughout the team, which can damage cooperation sustainability [6,14,15].

Team identity has been negatively related to free-riding behaviors. Team identity has been highlighted as an informal control system that refers to the internalized sense of individuals to be members of a group and their tendency to define themselves in terms of "we" rather than "I" [40,41]. An agent who identifies as a member of a team undergoes a psychological process, whereby s/he begins to think that s/he represents the group rather than acting as a unique individual [19,35]. This psychological process influences individuals' behavior. Agents wish to achieve group goals more than individual goals [19,42]. Therefore, since individuals who identify strongly with their team will focus on team goals instead of individual goals, they will cooperate more than those who do not identify with their team, because they wish to maximize group goals [19–22].

This previous reasoning can explain team identity positively responses to the greed component of social dilemmas. Greed is the temptation to behave as a free-rider when others cooperate. Free-riding behaviors in this situation represents the pure selfish behavior. If team identity increases the sense of

group, further cooperation is expected because the agent defines himself or herself in terms of "we" rather than "I", then less selfish behaviors are expected. However, free-riding behaviors in the fear component of social dilemmas, not necessarily represents selfish behaviors, but a response to fear. An agent can suffer fear because other team members are not contributing and are taking advantage from the agent´s contribution. In this situation, if an agent decides to contribute to the team, despite the teammate´s free-riding behaviors, he will be trying to maximize group goals, but not avoiding being exploited. Then, one way of reducing fear is acting as his/her teammates and free-ride. The question is whether team identity can explain agents' response to fear where two goals are opposed: maximizing ingroup goals and avoiding being exploited [9]. While some authors suggest that team identity focuses agents' behavior on the maximization of group goals, therefore, it is expected that agents will cooperate and will sacrifice. Others suggest that both goals are important when team identity is present; therefore, it is not clear how they will react when suffering fear (cooperating or free-riding) [9,43].

Trying to understand different agents' reactions to fear, we follow studies that suggest that women and men react differently to team identity in social dilemma contexts. Brown-Kruse and Hummels [25] found that men react more strongly to social identity than women in a multiperiod public good provision game. That is, men cooperate more than women, when social identity is present. In contrast, and replicating Brown-Kruse and Hummels [25], Cadsby and Maynes [28] found no differences according to gender. They only found an interaction effect between gender and social identity in the first round of the game, but in the opposite direction than Brown-Kruse and Hummels [25]: women contribute significantly more than men, but significance vanished as the game evolved. The design of these previous studies, however, does not facilitate in the same way the detection of free-riding among teammates and, thus, the appearance of fear. The design by Brown-Kruse and Hummels [25] facilitates the appearance of fear. Agents played a public goods game and had to make a binary decision (full or nothing). Moreover, they received feedback information, which facilitates the detection of free-riding teammates behavior. In contrast, in Cadsby and Maynes [28] agents had to make a continuous decision, being more difficult to detect free-riding behaviors. Following previous research, the present study wants to contribute to two lines of research. First, to the debate about the effect of team identity on free-riding reactions to fear. Second, to the debate about the interaction effect between team identity and gender. We combine team identity theory and gender literature to analyze the interaction effect between gender and team identity on free-riding behaviors because of fear, and then, on cooperation sustainability.

## 2.2. Hypotheses Development

Experimental evidence has extensively studied three different domains in which gender bias is present: social preferences, risk preferences, and reaction to competition [33,34]. Related to social preferences, women are more inequity averse and more affected by reciprocity behaviors than men [34,44,45]. Due to differences in social preferences, we suggest that men and women might react differently when suffering fear because of detecting teammates´ free-riding behaviors. For women it is important to avoid fear, and then reducing inequities in the group is key. However, men are less concerned about inequities and more concerned with maximizing outcomes [34,44]. Consequently, we might expect that the positive effect of team identity when agents suffer fear will be greater among men than women. That is, when team identity is high, men will decrease their level of free-riding responses to fear more than women.

Previous studies have also found that women are more risk averse than men [46]. To cooperate when other team members might free-ride could be perceived as a risky choice. Therefore, we suggest that women will be more motivated to free-ride than men when they are exposed to situations where teammates can free-ride due to their risk aversion. Finally, findings from economic literature suggest that women are more reluctant than men to engage in competitive interactions (tournaments, bargaining, or auctions) [34]. The setting analyzed in this study facilitates within-group competition due to the feedback information shared among teammates. In this context, if agents cooperate and one

teammate decides to free-ride, agents might feel that they are playing a sucker role. In this situation, agents might be motivated to defect in order to punish their teammates and avoid feeling like losers within the group [11,12,47]. Therefore, we might expect more free-riding behaviors among men than women, due to competition motives. However, previous studies suggest that team identity eliminates the sense of competition within the group (ingroup competition) and shifts the focus of the competition outside the group (outgroup competition) [20]. Therefore, we propose that the expected effect of competition on men's behavior (e.g., increasing free-riding responses) will not be present when team identity is strong.

In summary, due to differences between men and women in terms of inequity, risk preferences, and reaction to competition, we suggest that men and women's reactions to fear in social dilemma contexts will be different when team identity is strong. Overall, the previous line of reasoning suggests that team identity will affect free-riding responses to fear to a lesser extent in women than in men. That is, when team identity is strong, we might expect men to decrease their level of free-riding responses to fear more than women. In this respect, we expect results in line with those of Brown-Kruse and Hummels [25], and Solow and Kirkwood [26], but which counter the findings of Cadsby and Maynes [28]. The explanation is related to the setting of the different studies. In the former, the setting facilitates detection of free-riding behaviors, even though these studies did not focus on free-riding responses to fear [25,26]. In contrast, in the study conducted by Cadsby and Maynes [28], no detection of free-riding was possible. Therefore, we propose that in work contexts where feedback information facilitates the detection of free-riding behaviors, men will react more strongly than women to team identity by decreasing their level of free-riding responses.

Considering that sustainability refers to the capability of teammates to maintain cooperation, and considering that increasing the level of free-riding responses to fear will decrease cooperation along periods, we expect that the effect of team identity on cooperation sustainability will be higher among men than women. Accordingly, we formulate two hypotheses:

**Hypothesis 1 (H1).** *The effect of team identity on free-riding responses to fear will be higher among men than women.*

**Hypothesis 2 (H2).** *The effect of team identity on cooperation sustainability will be higher among men than women.*

We summarize our model in Figure 1.

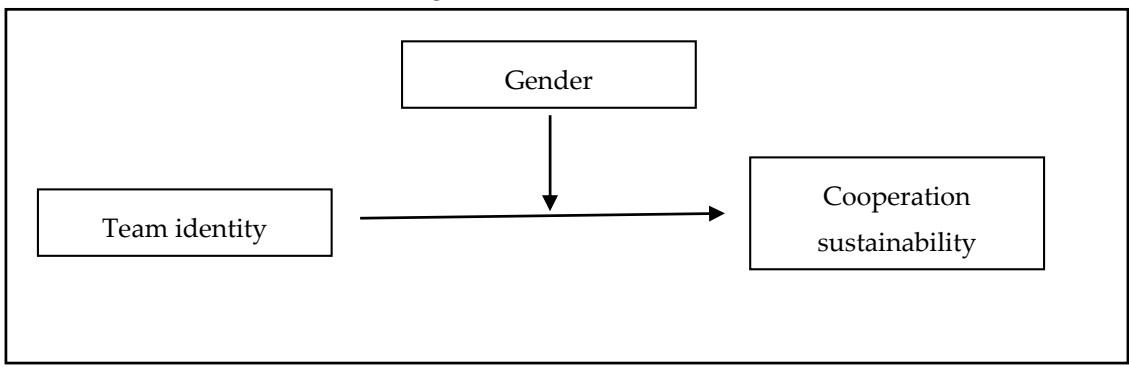

**Figure 1.** Model of the study.

## 3. Experimental Design

### 3.1. Experimental Procedures and Task Description

To test our hypotheses, we designed a 2 × 2 experiment in which we manipulated the level of team identity and controlled for gender differences. Experiments are a useful and appropriate mechanism

for analyzing causal relations under pure and uncontaminated conditions [48]. That is, the use of experiments as research methodology allows isolating in an artificial way the situation to analyze.

The task encompassed 30 periods, and each period was a within-subjects factor. The task represents a prisoner's dilemma game, which has been extensively used to represent social dilemmas within organizations [6,19,42,49]. Neoclassical Economic theory predicts that agents will behave as homo economicus, that is, agents will free-ride to gain benefits for the time, money, or effort of their teammates. However, from a psychological point of view, researchers suggest that agents do not behave rationally and self-interested, but are influenced by social aspects. It is extensively used social dilemmas situations, as the prisoner's dilemma, in management accounting, and economic literature, to show that people not always behave rationally, but they are influenced by social aspects, such as, social comparison, honesty, or team identity, e.g., [6,19,50,51].

To facilitate the appearance of fear, direct reciprocity was facilitated by providing feedback information after each period. Participants were given information about the contribution and earnings of each teammate for the just-finished period. A total of 84 graduate students participated in the experiment; 46.43% were male and 53.57% female. Students were adequate since no specific knowledge or previous experience was needed to perform the experimental tasks [52].

Experimental sessions were conducted in 2012 in a laboratory and lasted approximately 50 min. Separate sessions were conducted for each condition. Instructions were read aloud to the participants by one of the authors; then, instructions were given individually to all participants. The same written instructions were used in all four experimental conditions, to avoid framing effects from ex-ante information. To ensure that participants understood the instructions, they were required to score 100% on a computerized pre-experiment quiz before beginning the actual experiment (see Appendix A). Teammates could not communicate with each other during the task and sat in different spaces.

The experimental task was designed using Z-Tree software [53] and was adapted to three-person teams following Coletti et al. [6] and Gomez-Ruiz [49]. Prisoner´s dilemma games are extensively used in experimental economics. Research can use the same setting to test different models and hypothesis. Using the same experimental setting, in the present paper, independent variables are team identity and gender, and the dependent variables free-riding responses to fear and cooperation sustainability. While in Gómez-Ruiz (2015) the independent variable was the design of performance reports, the dependent variable was the total level of cooperation and the mediating variable was social comparison.

A total of 28 teams participated in the task. Participants were assigned to a three-person team, with each participant representing an independent observation. While each participant had the same partners throughout the experimental session, each person knew only the partners' participant numbers, not their identities.

Participants assumed the role of a research and development (R&D) manager at a pharmaceutical company. The company asked the three R&D managers to work together on a team project to develop a new product. For each period, each manager had to decide how much of his or her division's R&D resources to devote to the joint project. There were only two choices: a high or a low level of resources. Each participant incurred a cost of 15 points if s/he decided to invest a high level of resources. The cost was zero points if the participant chose a low level of resources. Since the three participants equally shared the income from the joint project, this cost provided an incentive for the participants to engage in free-riding—that is, to devote only a low level of resources to the joint project. Participants made their decisions for 30 separate periods. Experimental parameters are summarized in Exhibit 1 and Exhibit 2.

Exhibit 2 shows a prisoner's dilemma situation, which is created by the design of the team incentive [6,19]. As we define a three-player repeated game, we represent Player 1 in the row, Player 2 in the column, and Player 3 in the matrix [54]. The dilemma is that team performance is higher and everyone may benefit if all agents choose high levels of resources (cell 10, 10, 10). This option is the Pareto optimal solution. However, at the margin, each agent is better off free-riding, that is, choosing

low levels of resources (cell 5, 5, 5). In this prisoner's dilemma, the unique Nash equilibrium is for each agent to devote low resources to the joint project (cell 5, 5, 5). Figure 2 shows the prisoner's dilemma situation for a one-period decision, but the Nash equilibrium is also the solution in multiperiod decisions when agents know the number of times, they will play the game [55,56].

---

**Exhibit 1**

*Earning structure of the experiment (based on Coletti et al. [6] and Gomez-Ruiz [49])*

N = {1, 2, 3} (participants/team).

Binary decisions

H: high level of resources (cost = 15)
L: low level of resources (cost = 0)

Joint project income is shared equally by the three team members (1/3).

Joint project income increases with the level of resources dedicated to the joint project:

If three R&D managers choose low (L, L, L)

Project income = 15 points; $\prod$ = 15/3 = 5

If three R&D managers choose high (H, H, H)

Project income = 75 points; $\prod$ = 75/3 = 25

If any other situation

Project income = 45 points; $\prod$ = 45/3 = 15

(H, L, L) (H, H, L) (H, L, H) (L, H, L) (L, H, H) (L, L, H)

---

**Exhibit 2. Strategic form for three-player game.**

|  |  | Player 3: High | | Player 3: Low | |
|---|---|---|---|---|---|
|  |  | Player 2 | | Player 2 | |
|  |  | High | Low | High | Low |
| Player 1 | High * | 10, 10, 10 ** | 0, 15, 0 | 0, 0, 15 | 0, 15, 15 |
|  | Low * | 15, 0, 0 | 15, 15, 0 | 15, 0, 15 | 5, 5, 5 |

\* Represents the level of resources dedicated to the joint project (high or low).

\*\* Represents the earning (points) to Player 1, Player 2, and Player 3, respectively, for each period decision.

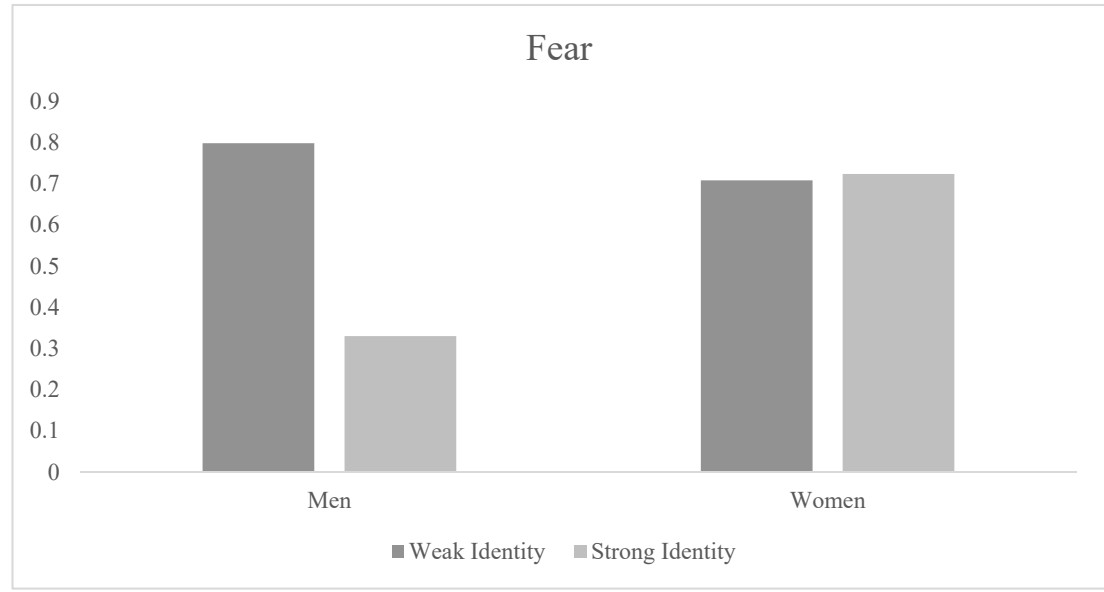

**Figure 2.** Fear across all four conditions (N = 84).

Participants were informed that they would receive a reward in real money, which depended on their department's profit at the end of the task (1 euro for every 50 points of profit earned) [6].

The department profit was calculated period by period and was the difference between 1/3 of the project income and the department's cost. Participants could receive between zero and nine euros for their participation. The average total earnings were 4.81 euros.

*3.2. Manipulation and Measurement*

The design fully crosses the measured independent variable of participant gender (men vs. women) with the manipulated independent variable of team identity. We did not create all-male or all-female groups, which might signal to the participants "that gender was a subject of study and lead them to alter their behavior" [26] (p. 405). Team identity was manipulated through the salience dimension [19]. We combined two procedures used in previous studies: color T-shirts and competition between teams. First, participants who worked for the same company wore the same color T-shirt to increase their level of team identity [19,20,22]. The strong team identity condition always had at least two groups of participants with different color T-shirts [19]. Second, participants in the strong team identity condition were informed that the companies they represented were competitors, and that the new R&D project was very important for increasing market share, although this competition did not change the team incentive across conditions. In the weak team identity condition, participants wore their own clothes and were not informed about any market share competition between companies. We checked if team identity was adequately manipulated by the item 16 included in post questionnaire (see Appendix B). Individuals who wore the same color T-shirt and were informed that the companies they represented were competitors scored higher in the item 16 than those who wore a different color T-shirt (F = 6.265, *p*-value = 0.014).

The first dependent variable was free-riding responses to fear. We called it fear variable and it represents the number of times an agent chooses to free-ride after a period in which their two teammates free-rode. The second dependent variable was cooperation sustainability, which depends on the cooperation variable. First, when an agent chose a low level of resources in a period, we coded this variable as "0"; conversely, if he or she chose a high level of resources, we coded this variable as "1". Cooperation was the sum of resources chosen by each agent over the 30 periods (range 0–30), transformed into a ratio (0 = 0 percent to 30 = 100 percent cooperation). Then cooperation sustainability was determined by the extent to which cooperation declined over periods, following Kelly and Tan [7]. Sustainability of cooperation variable was measured through the difference between the mean cooperation for the last five periods and the mean for the first five periods, transformed into a ratio (percent). A negative ratio means that cooperation is lower in the last periods than in the first periods (i.e., cooperation is not sustained over periods). A positive ratio means that cooperation is greater in the last periods than in the first periods (cooperation is sustained over periods).

We also measured early, middle, and late cooperation, and early, middle, and late fear variables. These variables are explained in more detail in the Results section. Finally, at the end of the task, participants responded to a questionnaire to verify the manipulation of the variables and their understanding of the procedures (see Appendix B).

## 4. Results

*4.1. Descriptive Statistics*

Table 1 reports descriptive statistics for fear, early fear, middle fear, and late fear. Table 1 also reports cooperation sustainability, cooperation, early cooperation, middle cooperation, and late cooperation. Figures 2 and 3 report fear and cooperation sustainability variables for all four conditions.

**Table 1.** Descriptive statistic: Mean {SD}.

| | Weak-Identity Teams | | Strong-Identity Teams | |
|---|---|---|---|---|
| | Gender | | Gender | |
| | Men (N = 16) | Women (N = 26) | Men (N = 23) | Women (N = 19) |
| Fear [a] | 0.798 {0.170} | 0.708 {0.280} | 0.330 {0.378} | 0.724 {0.230} |
| Early Fear [b] | 0.765 {0.363} | 0.593 {0.432} | 0.270 {0.315} | 0.519 {0.368} |
| Middle Fear [c] | 0.871 {0.132} | 0.691 {0.326} | 0.403 {0.455} | 0.594 {0.347} |
| Late Fear [d] | 0.843 {0.188} | 0.725 {0.302} | 0.429 {0.432} | 0.835 {0.222} |
| Cooperation Sustainability [e] | −0.579 {0.580} | −0.537 {0.533} | −0.241 {0.557} | −0.390 {0.588} |
| Cooperation f | 0.460 {0.317} | 0.536 {0.266} | 0.687 {0.302} | 0.598 {0.253} |
| Early Cooperation [g] | 0.644 {0.303} | 0.715 {0.280} | 0.717 {0.261} | 0.663 {0.211} |
| Middle Cooperation [h] | 0.450 {0.403} | 0.515 {0.299} | 0.748 {0.337} | 0.674 {0.296} |
| Late Cooperation [i] | 0.288 {0.320} | 0.377 {0.365} | 0.596 {0.401} | 0.458 {0.322} |

[a] Fear: the denominator is the total number of times an agent faces a situation where their two teammates free-ride in the previous round. The numerator is equal to the number of times the agent chooses to free-ride in the following round. [b] Early Fear: the denominator is the total number of times an agent faces a situation where their two teammates free-ride in the previous round over the 10 first periods. The numerator is equal to the number of times the agent chooses to free-ride in the following round. [c] Middle Fear: the denominator is the total number of times an agent faces a situation where their two teammates free-ride in the previous round between periods 11 and 20. The numerator is equal to the number of times the agent chooses to free-ride in the following round. [d] Late Fear: the denominator is the total number of times an agent faces a situation where their two teammates free-ride in the previous round between periods 21 and 20. The numerator is equal to the number of times the agent chooses to free-ride in the following round. [e] Cooperation sustainability: the difference between the mean cooperation for the last five periods and the mean for the first five periods, transformed into a ratio (percent). [f] Cooperation: The sum of resources chosen by each agent over the 30 periods (range 0–30), transformed into a ratio (0 = 0 percent to 30 = 100 percent cooperation): [g] Early cooperation: the sum of resources chosen by each agent over the 10 first periods (range 0–10), transformed into a ratio (0 = 0 percent to 10 = 100 percent cooperation). [h] Middle cooperation: the sum of resources chosen by each agent between periods 11 and 20 (range 0–10), transformed into a ratio (0 = 0 percent to 10 = 100 percent cooperation). [i] Late cooperation: the sum of resources chosen by each agent between periods 21 and 30 (range 0–10), transformed into a ratio (0 = 0 percent to 10 = 100 percent cooperation).

The fear variable represents the number of times an agent free-rides following a round in which their two teammates free-rode. The highest level of fear response is found among men and in the weak team identity condition (M = 0.798, SD = 0.170), followed by women and the strong team identity condition (M = 0.724, SD = 0.230), then women and the weak team identity condition (M = 0.708, SD = 0.280). The lowest level of fear is found among men and in the strong team identity condition (M = 0.330; SD = 0.378). Related to early, middle, and late fear, the highest level of early fear is found among men and in the weak team identity condition (M = 0.765, SD = 0.363), followed by women and the weak team identity condition (M = 0.593, SD = 0.432), then women and the strong team identity condition (M = 0.519, SD = 0.368). The lowest level of early fear is found among men and in the strong team identity condition (M = 0.270; SD = 0.315). The highest level of middle fear is found among men and in the weak team identity condition (M = 0.871, SD = 0.132), followed by women and the weak team identity condition (M = 0.691, SD = 0.326) and then women and the strong team identity condition (M = 0.594, SD = 0.347). The lowest level of middle fear is found among men and in the strong team identity condition (M = 0.403; SD = 0.455). Finally, similar levels of late fear were found among men and in the weak team identity condition (M = 0.843; SD = 0.188) and women and the strong team identity condition (M = 0.835, SD = 0.222). The lowest level of late fear was found among

men and in the strong team identity condition (M = 0.429; SD = 0.432), followed by women and the weak team identity condition (M = 0.725, SD = 0.302).

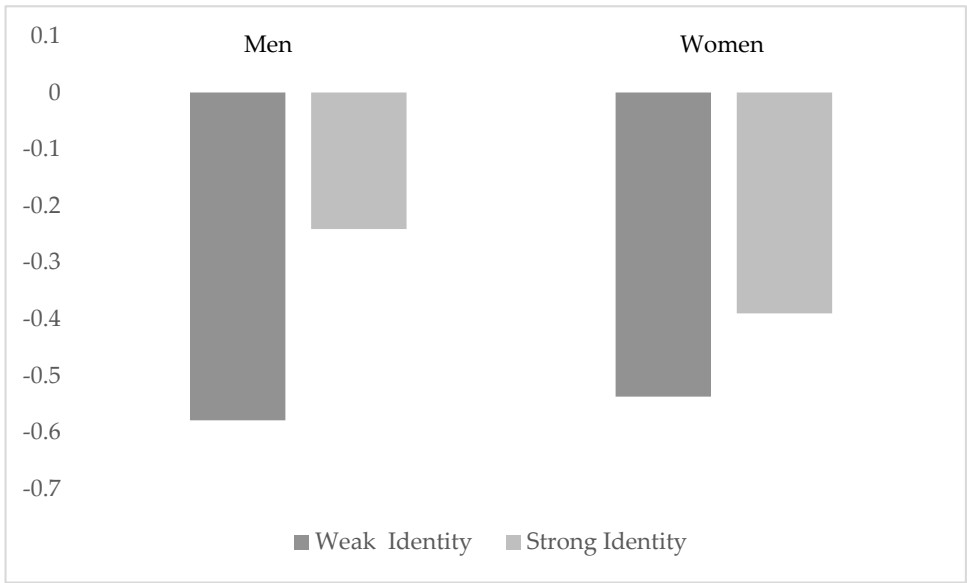

**Figure 3.** Cooperation sustainability across all four conditions (N = 84).

The cooperation variable measures the extent to which agents choose a high level of resources (i.e., cooperation) instead of a low level of resources (i.e., free-riding) over different periods. The lowest level of cooperation is found among men and in the weak team identity condition (M = 0.460, SD = 0.317), followed by women and the weak team identity condition (M = 0.536, SD = 0.266), and women and the strong team identity condition (M = 0.598; SD = 0.253). The highest level of cooperation is found among men and in the strong team identity condition (M = 0.687, SD = 0.302). The early cooperation variable measures the extent to which participants choose a high level of resources instead of a low level of resources over the 10 initial periods. The lowest mean for early cooperation is found among men and in the weak team identity condition (M = 0.644, SD = 0.303), followed by women and the strong team identity condition (M = 0.663, SD = 0.211). Similar levels of early cooperation are found among women and the weak team identity condition, and men and the strong team identity condition (M = 0.715, SD = 0.280; M = 717, 0.261, respectively). The middle cooperation variable measures the extent to which participants choose a high level of resources instead of a low level of resources between periods 11 and 20. The lowest mean for middle cooperation is found among men and in the weak team identity condition (M = 0.450, SD = 0.403), followed by women and the weak team identity condition (M = 0.515, SD = 0.299), and then women and the strong team identity condition (M = 0.674, SD = 0.296). The highest level of cooperation was found among men and in the strong team identity condition (M = 0.748, SD = 0.337). The late cooperation variable measures the extent to which participants choose a high level of resources instead of a low level of resources between periods 21 and 30. The lowest mean for late cooperation is found among men and in the weak team identity condition (M = 0.288, SD = 0.320), followed by women and the weak team identity condition (M = 0.377, SD = 0.365), and then women and the strong team identity condition (M = 0.458, SD = 0.322). The highest level of cooperation was found among men and in the strong team identity condition (M = 0.596, SD = 0.401).

Finally, cooperation sustainability represents the tendency along periods. A negative ratio means that cooperation is lower in last periods than in first periods. We found a negative ratio among the four conditions. However, the lowest negative ratio is found among men and in the strong team identity condition (M = −0.241, SD = 0.557), followed by women and the strong team identity condition (M = −0.390, SD = 0.588), and women and the weak team identity condition (M = −0.537; SD = 0.533).

The highest negative ratio of cooperation sustainability is found among men and in the weak team identity condition (M = −0.579, SD = 0.580).

*4.2. Hypotheses Testing*

Hypothesis 1 stated that the effect of team identity on free-riding response to fear would be lower among women than men. We used an ANOVA model in which the dependent variable was fear variable, and the independent variables were gender and team identity. Results in Table 2 showed a statistically significant and positive effect for team identity on fear (F = 10.091, *p* = 0.002), a statistically significant effect for gender on fear (F = 4.527, *p* = 0.038), and a statistically significant interaction effect (team identity × gender) on fear (F = 11.517; *p* = 0.001). Therefore, hypothesis 1 is supported.

**Table 2.** ANOVA model (N = 84). Dependent variable: fear.

|  | Sum of Squares | DF | Mean Square | F | *p*-Value |
|---|---|---|---|---|---|
| Team identity | 0.772 | 1 | 0.772 | 10.091 | 0.002 *** |
| Gender | 0.346 | 1 | 0.346 | 4.527 | 0.038 ** |
| Team identity x gender | 0.881 | 1 | 0.881 | 11.517 | 0.001 *** |

**, *** are significant at 5% and 1%, respectively (two-tailed).

In order to test the robustness of our results, we also analyzed hypothesis 1 with simple effect analysis (see Table 3). Results showed no significant difference in free-riding responses to fear for women, between the weak and strong team identity conditions (t = −0.168; *p* = 0.868). Results showed significant differences in free-riding response to fear for men, between the weak and strong team identity conditions (t = 4.229; *p* < 000). As expected, women developed similar levels of free-riding response to fear indistinctly of team identity strength (weak vs. strong: 0.708 vs. 0.724), while men developed different levels of free-riding response to fear depending on the strength of team identity (weak vs. strong: 0.798 vs. 0.330). Therefore, simple analysis results are in line with the findings of the ANOVA model, supporting Hypothesis 1.

**Table 3.** Simple effect analysis (N = 84). Dependent variable: fear.

|  | Weak | Strong | DF | T-Stat | *p*-Value |
|---|---|---|---|---|---|
| Effect of team identity on fear responses for men | 0.798 | 0.330 | 26 | 4.229 | <0.000 *** |
| Effect of team identity on fear responses for women | 0.708 | 0.724 | 33 | −0.168 | 0.868 |

*** is significant at 1% (two-tailed).

Hypothesis 2 stated that the effect of team identity on cooperation sustainability would be higher among men than women. We developed two analyses. First, we used an ANOVA model in which the dependent variable was cooperation, and the independent variables were gender and team identity. Results in Table 4 showed only a statistically significant and positive effect for team identity on cooperation (F = 5.265, *p* = 0.024). However, no statistically significant effect was found for gender (F = 0.011; *p* = 0.917), and no interaction effect (team identity x gender) was observed on cooperation (F = 1.701; *p* = 0.196). Furthermore, we developed a simple effect analysis (see Table 5). Results showed no significant difference in cooperation for women, between the weak and strong team identity conditions (t = −0.800; *p* = 0.428). Results showed significant differences in cooperation for men, between weak team identity and low team identity (t = −2.237; *p* = 0.033). As expected, women developed similar levels of cooperation indistinctly of team identity strength (weak vs. strong: 0.536 vs. 0.598), while men developed different levels of fear responses depending on the strength of team identity (weak vs. strong: 0.460 vs. 0.687).

**Table 4.** ANOVA model (N = 84). Dependent variable: cooperation.

|  | Sum of Squares | DF | Mean Square | F | *p*-Value |
|---|---|---|---|---|---|
| Team identity | 0.423 | 1 | 0.423 | 5.265 | 0.024 ** |
| Gender | 0.001 | 1 | 0.001 | 0.011 | 0.917 |
| Team identity x Gender | 0.137 | 1 | 0.137 | 1.701 | 0.196 |

** is significant at 5% (two-tailed).

**Table 5.** Simple effect analysis (N = 84). Dependent variable: cooperation.

|  | Weak | Strong | DF | T-Stat | *p*-Value |
|---|---|---|---|---|---|
| Effect of team identity on fear responses for men | 0.460 | 0.687 | 31.425 | −2.237 | 0.033 ** |
| Effect of team identity on fear responses for women | 0.536 | 0.598 | 40.048 | −0.800 | 0.428 |

** is significant at 5% (two-tailed).

We analyzed the effect of team identity and gender on cooperation sustainability, and results are in line with previous results about cooperation variable. We used an ANOVA model in which the dependent variable was cooperation sustainability, and the independent variables were gender and team identity. Results in Table 6 show only a statistically marginally significant and positive effect for team identity on cooperation sustainability (F = 3.776, $p$ = 0.056). However, no statistically significant effect was found for gender (F = 0.182; $p$ = 0.670), and no interaction effect (team identity x gender) was observed on cooperation (F = 0.592; $p$ = 0.444). Therefore, this ANOVA analysis did not support hypothesis 2. We also developed a simple effect analysis (see Table 7). Results showed no significant difference in cooperation sustainability for women, between the weak and strong team identity conditions (t = −0.856; $p$ = 0.397). Results showed marginally significant differences in cooperation for men, between weak team identity and low team identity (t = −1.820; $p$ = 0.078). Therefore, simple effect analysis support hypothesis 2, for cooperation variable and marginally for cooperation sustainability variable.

**Table 6.** ANOVA model (N = 84). Dependent variable: cooperation sustainability.

|  | Sum of Squares | DF | Mean Square | F | *p*-Value |
|---|---|---|---|---|---|
| Team identity | 01.189 | 1 | 1.189 | 3.776 | 0.056 * |
| Gender | 0.057 | 1 | 0.057 | 0.182 | 0.670 |
| Team identity x gender | 0.186 | 1 | 0.186 | 0.592 | 0.444 |

* is significant at 10% (two-tailed).

**Table 7.** Simple effect analysis (N = 84). Dependent variable: cooperation sustainability.

|  | Weak | Strong | DF | T-Stat | *p*-Value |
|---|---|---|---|---|---|
| Effect of team identity on fear responses for men | −0.579 | −0.241 | 31.568 | −1.820 | 0.078 * |
| Effect of team identity on fear responses for women | −0.537 | −0.390 | 36.624 | −0.856 | 0.397 |

* is significant at 10% (two-tailed).

*4.3. Supplemental Analysis*

We used early, middle, and late fear and cooperation variables, and post-experiment questionnaire data to provide support to the theory underlying our results.

4.3.1. Time Effect: Early, Middle and Late Fear Responses

To further develop our hypothesis 1, we ran pairwise comparisons between conditions (*t*-test analysis) for early fear, middle fear, and late fear variables. First, among men, between the weak and strong team identity conditions (Table 8a), we found statistically significant differences for early, middle, and late fear (t = 3.345, *p* =0.003; t = 3.107, *p* = 0.008; t = 3.235, *p* = 0.004). However, among women (Table 9b), we found no statistically significant differences for early fear, middle fear, or late fear (t = 0.435, *p* =0.669; t = 0.673, *p* = 0.513; t = −1.176, *p* = 0.249). Therefore, these results are in line with previous hypothesis 1 analysis. Figures 4 and 5 report the evolution of free-riding response to fear (early, middle, and late) for men and women, respectively, depending on their strength of team identity.

**Table 8.** Results from *t*-tests for early, middle, and late fear.

(**a**) Men condition, weak-identity teams vs. strong-identity teams (N = 39).

|  | **Weak** | **Strong** | **DF** | **T-Stat** | ***p*-Value** |
|---|---|---|---|---|---|
| Early fear | 0.765 | 0.270 | 18.069 | 3.345 | 0.003 *** |
| Middle fear | 0.870 | 0.403 | 14 | 3.107 | 0.008 *** |
| Late fear | 0.843 | 0.429 | 23 | 3.235 | 0.004 *** |

(**b**) Women condition, weak-identity teams vs. strong-identity teams (N = 45).

|  | **Weak** | **Strong** | **DF** | **T-Stat** | ***p*-Value** |
|---|---|---|---|---|---|
| Early fear | 0.593 | 0.519 | 19.108 | 0.435 | 0.669 |
| Middle fear | 0.690 | 0.594 | 12.484 | 0.670 | 0.513 |
| Late fear | 0.725 | 0.835 | 28.593 | −1.176 | 0.249 |

*** is significant at 1% (two-tailed).

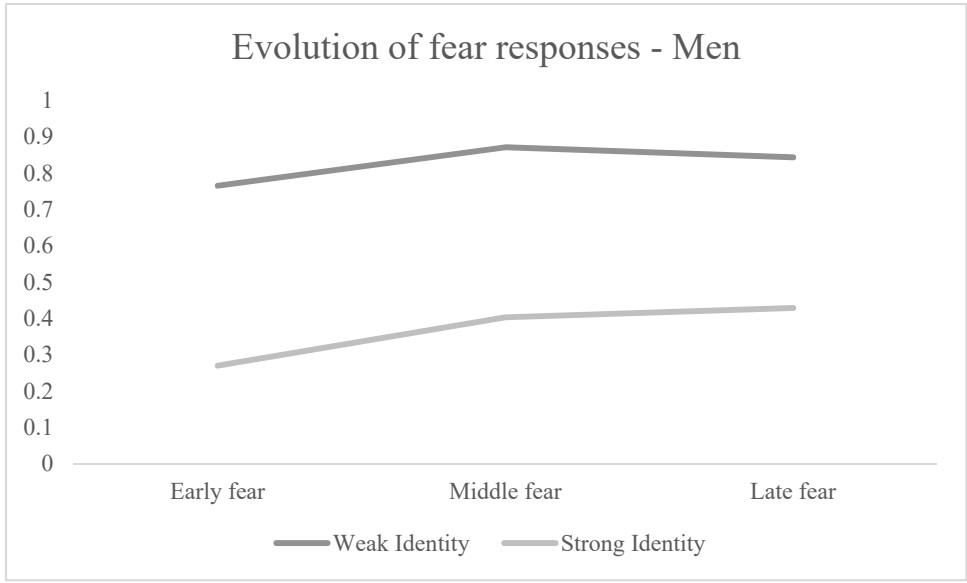

**Figure 4.** Evolution of fear within Men condition (N = 39).

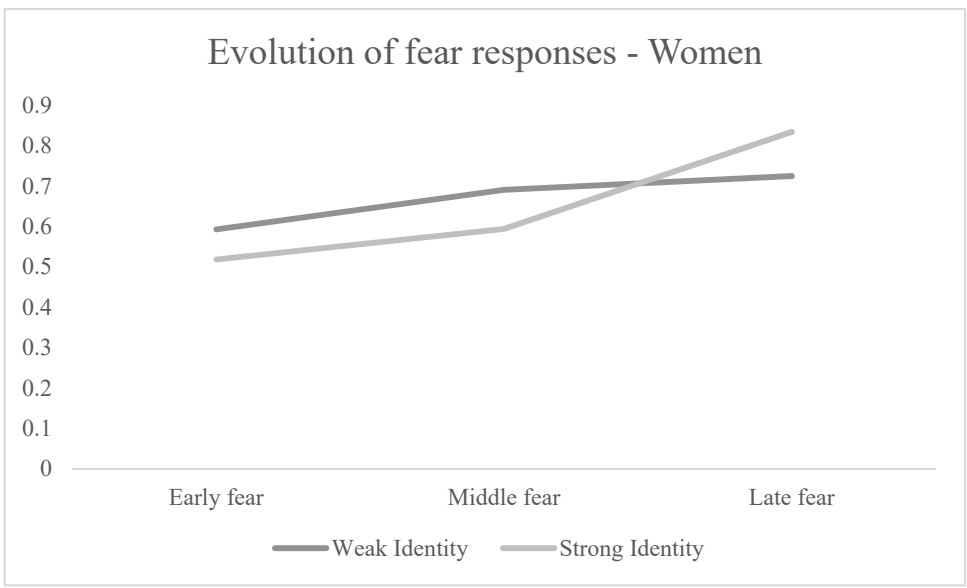

**Figure 5.** Evolution of fear within Women condition (N = 45).

### 4.3.2. Time Effect: Early, Middle, and Late Cooperation

We ran pairwise comparisons between conditions (*t*-test analysis) for early cooperation, middle cooperation, and late cooperation variables, to examine the effect of team identity along periods, that is, to give greater support to hypothesis 2. First, among men, between the weak and strong team identity conditions (Table 9a), we found no statistically significant differences for early cooperation (t = −0.79, *p* = 0.436). However, we found statistically significant differences for middle and late cooperation (t = 2.423, *p* = 0.022; t = −2.664, *p* = 0.011). Therefore, the results suggest that the effect of team identity on men is not automatic, but rather needs time in order to influence men's behavior. We suggest that this delay in influencing cooperative behaviors among men might explain the non significance of the ANOVA model with regard to supporting hypothesis 2. However, among women (Table 9b), we found no statistically significant differences for early, middle, or late fear (t = 0.714, *p* = 0.479; t = −1.765, *p* = 0.085; t = −0.787, *p* = 0.436). Therefore, results support the positive effect of team identity on cooperation sustainability for men but not for women.

**Table 9.** Results from *t*-tests for early, middle, and late cooperation.

(**a**) Men condition, weak-identity teams vs. strong-identity teams (N = 39).

|  | Weak | Strong | DF | T-Stat | *p*-Value |
|---|---|---|---|---|---|
| Early fear | 0.643 | 0.717 | 29.122 | −0.790 | 0.436 |
| Middle fear | 0.450 | 0.748 | 28.517 | −2.423 | 0.022 ** |
| Late fear | 0.288 | 0.596 | 36.193 | −2.664 | 0.011 ** |

(**b**) Women condition, weak-identity teams vs. strong-identity teams (N = 45).

|  | Weak | Strong | DF | T-Stat | *p*-Value |
|---|---|---|---|---|---|
| Early fear | 0.715 | 0.663 | 42.927 | 0.714 | 0.479 |
| Middle fear | 0.515 | 0.674 | 39.167 | −1.765 | 0.085 |
| Late fear | 0.377 | 0.458 | 41.385 | −0.787 | 0.436 |

** is significant at 5% (two-tailed).

### 4.3.3. Post-Experiment Questionnaire Data

We used post-experiment questionnaire data to support the theory underlying free-riding response to fear and differences between men and women. We used two questions in the post-experiment questionnaire (see Appendix C) to capture the frequency with which agents think about following teammates' free-riding behaviors, and, therefore, the frequency with which agents think about engaging in punishment behaviors (free-riding) in response to fear. We developed a simple effect analysis and compared the responses of men and women within the weak and strong team identity conditions.

First, we ran pairwise comparisons between men and women (*t*-test analysis) within the weak team identity condition, for the two items (Table 10a). We found no statistically significant differences for item 1 and item 2, between men and women (t = −0.743, *p* =.462; t = −0.741, *p* =.464). Therefore, men and women think about developing similar levels of punishment when they work in teams with a weak identity. Second, we ran pairwise comparisons between men and women (*t*-test analysis) within the strong team identity condition, for the two items. We found statistically significant differences for item 1 (t = −2.032, *p* =0.049) and marginally statistically significant differences for item 2 (t = −1.807, *p* =0.079). Therefore, team identity changes the thoughts of men and women related to punishing behaviors towards their teammates. Women think more about following free-riders than men (item 1: women 3.47 vs. men 2.65; item 2: women 3.26 vs. men 2.48).

**Table 10.** Results from *t*-tests for items 1 and 2 (Appendix C, Post-questionnaire for supporting fear variable).

(**a**) Weak-identity team condition, Men vs. Women (N = 42).

|  | **Men** | **Women** | **DF** | **T-Stat** | ***p*-Value** |
|---|---|---|---|---|---|
| Item 1 | 2.88 | 3.15 | 36.469 | −0.743 | 0.462 |
| Item 2 | 3.13 | 3.42 | 34.919 | −0.741 | 0.464 |

(**b**) Strong-identity team condition, Men vs. Women (N = 42).

|  | **Men** | **Women** | **DF** | **T-Stat** | ***p*-Value** |
|---|---|---|---|---|---|
| Item 1 | 2.65 | 3.47 | 38.449 | −2.032 | 0.049 ** |
| Item 2 | 2.48 | 3.26 | 37.300 | −1.807 | 0.079 * |

*, ** are significant at 10% and 5%, respectively (two-tailed).

## 5. Discussion

This study analyzed the interaction effect of gender and team identity on free-riding behaviors because of fear and cooperation sustainability. Untangling the link between gender, team identity, and economic behavior when agents face social dilemma situations is not straightforward [26]. Previous research found mixed results when analyzing the effects of team identity on free-riding response to fear [9,43]; however, we argue that this relationship depends on gender characteristics. Differences in three domains between men and women (inequity, risk preferences, and reaction to competition) [33,34] might explain the different effect of team identity on free-riding responses to fear and cooperation sustainability.

Our results show a significant interaction effect of gender and team identity on free-riding response to fear. Specifically, they show that the effect of team identity on free-riding response to fear is significant in the case of men, but not in the case of women. This result is consistent with our first hypothesis and extends the study by Simpson [9]. Simpson [9] argued that team identity does not affect fear responses because the decision to cooperate only achieves one of the two goals when team identity is strong. Cooperation when other team members free-ride maximizes group outcomes, but defection is what minimizes ingroup inequities. We found the effect suggested by Simpson [9] in women, but not in men. We provided different explanations for this different effect. For women it

is important to avoid fear and then reducing inequities in the group is key. Therefore, to decide to free-ride when agents are affected by fear situations minimizes ingroup inequities. Whereas for men, maximizing group outcomes is more important [34,44]. Second, previous findings suggest that men will be motivated to free-ride when others do so because of competitive motives, whereas women are more motivated by risk preferences. Team identity changes the focus of competition from within the group to outside the group, but it does not affect risk perception [20]. This suggests that team identity will affect decisions to cooperate when suffering fear in the case of men, but not in the case of women. Additionally, our results from the post-experimental questionnaire show that women and men decide to follow the free-riding behavior of other teammates to the same extent when team identity is weak. However, when team identity is strong, women are more willing to follow that behavior than men. These results are consistent with our expectations.

Considering that responding with free-riding behaviors to fear situations will negatively impact cooperation sustainability, we might expect an interaction effect for gender and team identity on cooperation sustainability. However, although the simple effect analysis shows that team identity only significantly affects cooperation in the case of men, and marginally affects cooperation sustainability, our results show that the interaction effect is not significant for cooperation or for cooperation sustainability. We provided an explanation related to time. The effect of team identity on cooperation among men is not immediate. As shown by our supplemental analysis, for men, the effect of team identity is significant in the last 20 periods, but not in the first 10. Another explanation might be that the effect of team identity on greed responses could compensate for the effect of team identity on fear responses. Future research could extend our study to other settings that boost the greed component of the social dilemma.

## 6. Conclusions

In work context where feedback information facilitates the detections of free-riding behaviors, our findings show no effect for team identity on free-riding response to fear among women. However, we found that team identity affects men's reaction by decreasing their level of free-riding responses to fear. Although considering that lower levels of free-rider responses to a fear situation will increase cooperation along periods, the results of this study suggest that the effect of team identity on cooperation among men is not immediate.

However, it should be noted that this study presents several limitations. Because of our experimental method, this study has limited generalizability to real-world settings where other variables may influence employee behaviors. In our experiment, team members could not interact or communicate. Although there are real work situations in which workers cannot interact with and monitor each other because of distance, for instance, they normally can communicate; and communication is positively related to cooperation [57]. Future research should test our predictions and seek to replicate our findings in real contexts. In addition, we did not have the opportunity to distinguish between the different motives that lead participants to defect (inequity preferences, risk preferences, or competition). Future research could analyze these mediating variables and other control variables that could also explain the differences found in our study.

This study has implications for social dilemma, gender, and informal control mechanism literature as well as practical implications for organizational sustainability. Considering that sustainability at the organizational level depends on the employee´s contribution to organizational and group goals, we contribute to answer how to create institutions that foster cooperative norms for disabling prisoner's dilemmas that are always part of business and society, by the use of informal control systems (e.g., team identity) [5]. In addition, we are contributing to the debate about the interaction between gender and team identity [27,37]. Our results are in line with Brown-Kruse and Hummels [25] and Solow and Kirkwood [26], but contrary to Cadsby and Maynes [28]. In the former, agents played a public goods game and had to make a binary decision (full or nothing), while, in the latter, agents had to make a continuous decision. Moreover, in the studies conducted by Brown-Kruse and Hummels [25] and Solow

and Kirkwood [26], no feedback was provided about teammates after each decision, but instead group information was made available. Similar feedback can be found in Cadsby and Maynes [28]. However, having a binary decision facilitates the interpretation of group feedback information and, therefore, facilitates the detection of free-riding. Along this line, we suggest that our setting is closer to that used by Brown-Kruse and Hummels [25] and Solow and Kirkwood [26] in their studies, and for this reason our results are in line, that is, men react more strongly to social identity than women. Furthermore, this study contributes to the debate about how team identity affects cooperation by extending the study of Simpson [9]. Simpson [9] suggests that team identity influences greed but not fear responses, because only when agents respond to the greed component with cooperation do they attain the two expected goals of social identity (maximizing group goals and reducing inequities). Our results show that team identity does not affect fear responses in the case of women, but it does affect them in the case of men. Finally, considering that the presence of women in organizations cannot be ignored [29,30,58], we have responded to recent calls to examine how new forms of work organization, such as team structures, and management control systems may adapt to gender differences [29]. We are contributing to this line of research by suggesting that team identity does affect men and women differently. Therefore, organizations should adapt managerial practices, formal and informal, according to gender.

**Author Contributions:** Conceptualization, L.G.-R. and M.J.S.-E.; methodology, L.G.-R. and M.J.S.-E.; formal analysis, L.G.-R. and M.J.S.-E.; investigation, L.G.-R. and M.J.S.-E.; data curation, L.G.-R. and M.J.S.-E; methodology, L.G.-R.; writing—original draft preparation, L.G.-R. and M.J.S.-E.; writing—review and editing, L.G.-R. and M.J.S.-E. All authors have read and agreed to the published version of the manuscript.

**Funding:** This research was funded by ANDALUSIAN REGIONAL GOVERMENT, grant number Project FEDER-UPO-1263739 and by Spanish Ministry of Education and Science, grant number Project PGC2018-094989-B-I00.

**Acknowledgments:** The authors express their thanks for comments received on earlier versions of this paper from David Naranjo-Gil, Natalia Jimenez and participants at Pablo Olavide University research seminars. The authors also acknowledge the partial funding of this research project by the Andalusia Regional Government (Project FEDER–UPO-1263739) and the Spanish Ministry of Education and Science (Project PGC2018-094989-B-I00).

**Conflicts of Interest:** The authors declare no conflict of interest.

## Appendix A. Pre-Questionnaire to Check Understanding of Instructions

In order to ensure that participants understood the guidelines for the activity, they had to answer the following questions:

1.  If you choose to invest a high level of resources in the joint project during a period, the cost for your division is:

    - 15 points
    - 0 points
    - 30 points

2.  If the three R&D managers choose a low level of resources for the joint project in a period, the income of the joint project is:

    - 0 points
    - 45 points
    - 15 points

3.  If the three R&D managers choose a high level of resources for the joint project in a period, the proportional share of the income of the joint project for every division is:

    - 25 points
    - 75 points
    - 45 points

4.　　If you have chosen to invest a high level of resources, but the other R&D manager(s) has (have) chosen a low level of resources, the income of the joint project is:

- 25 points
- 75 points
- 45 points

## Appendix B. Post-Questionnaire Containing the Manipulation Checks

Participants had to answer the items on a 5-point Likert scale ranging from 1 (completely disagree) to 5 (completely agree):

1.　　I have worked seriously on this task.
2.　　I was highly motivated to take part in this task.
3.　　Taking part in this task was fun.
4.　　The reward I receive from this task depends only on my decisions.
5.　　The common project's income depends on the decisions of every member in the team.
6.　　When taking part in the task, I felt that the three R&D managers set up a team.
7.　　I committed to my team during the task I performed.
8.　　My personal interests were not as important as the common interest of the team.
9.　　In any period of the task, the three team members could discuss our decisions.
10.　During the activity I could know who the members of my team were.
11.　I could meet the members of my team before the start of the task.
12.　I could meet the members of my team while I performed the task.
13.　After every decision, I knew the income of the common project.
14.　After every decision, I knew the reward of my own division.
15.　After every decision, I knew the reward of the two other divisions.
16.　Choose from the two figures below the one that best represents your feelings towards your team during the time that you have been participating in the study.

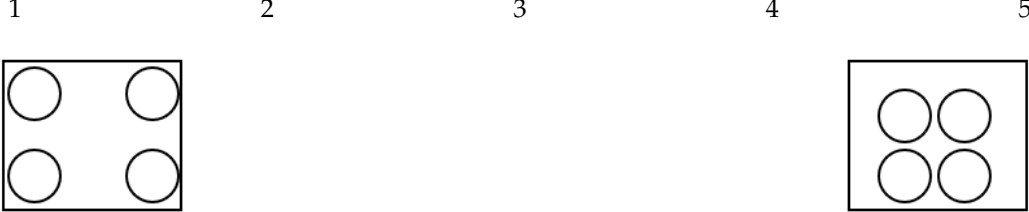

## Appendix C. Post-Questionnaire for Supporting Fear Responses

Participants had to answer the items on a 5-point Likert scale ranging from 1 (completely disagree) to 5 (completely agree):

There must have been situations in which you experienced someone investing lower than you in your team. In such a situation,

1.　　How often did you think, "That will happen to me, too"?
2.　　How often did you think, "Presently, I will do the same".

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
