# Peer review of "The Impact of Team Identity and Gender on Free-Riding Responses to Fear and Cooperation Sustainability"

_sustainability, doi:10.3390/su12198175_

Round 1

Reviewer 1 Report

I enjoyed reading your well-written article and agree that your research theme is interesting and potentially important. That said, I felt that some issues related to the current version limit its potential validity.

 I have outlined a number of points that should be addressed in any future version of this paper.

1 - Given the high number of variables mentioned in the study (Informal control system; gender; cooperation; cooperation sustainability; fear response; free-riding behaviors; team identity; social dilemma), it becomes difficult to understand the relationship between these variables and what is going to be studied. I had to read the article more than once to understand. I think that if the authors were able to present a Study research design this would be easier for the reader to understand what is being studied.

2 –  I think the title of the article is too broad and does not reflect what is studied. “The interaction between informal control systems and gender: Effects on fear responses and cooperation sustainability”.

In this study, only one informal control mechanism (team identity) was studied and not the entire informal control system of the company. Examples of informal management control mechanism are unwritten norms, loyalties, shared values, organizational culture and ethics, mutual commitments. Informal control includes high levels of professional and cultural control.

On the other hand, I think that what is studied is the impact of team-identity on self-interested behaviors (free-riding) and on cooperation sustainability.

3 - The discussion of the results and the conclusions sections should be separated

The 'Conclusion' section should provide a link and summary to the paper's introduction and objectives and should be structured as follows:

  • Brief summary of the paper's findings.
  • Limitations of the research and findings.
  • Implications for practitioners and researchers.
  • Possible areas for future research

4 - Least but not least, many similarities were found with previous works.

Line 195 – The experimental design is very similar to the study developed by:  Laura Gomez-Ruiz (2015) Performance reports and social comparison in group settings: effects on cooperation, Spanish Journal of Finance and Accounting / Revista Española de Financiación y Contabilidad, 44:1, 97-115, DOI: 10.1080/02102412.2014.996358

The differences between the two studies should be highlighted so that it is possible to better understand what are the diferences and contributions of each study.

Line 245 – Exhibit 2 is the same as Gomez-Ruiz (2015)

Reviewer 2 Report

Dear authors,

first of all, thank you for sharing your paper with me. I must admit that on a first read, I found the paper well written and interesting. On a second glimpse though, some questions and inconsistencies arose that need your attention. 

First of all, I am doubtful if it is justified to start with the hypothesis that men and women are necessarily firm categories that can be compared against eachother. At least, it would be necessary to speak of alternative categories that might be of equal importance. For recent literature on the subject, please see for example

Bagrationi K.Thurner T. Using the future time perspective to analyse resistance to, and readiness for, change // Employee Relations. 2020. Vol. 42. No. 1. P. 262-279. 

Secondly, I find the terms that you use not well justified. The term informal control system cannot be used interchangably with group identity. Indeed, there might be groups that punish diverting behavior, but there are also others that dont control or punish at all. You also introduce terms like trust and fear (in fear response). These are big terms with ample psychological research. More grounding in literature would be required.

Regarding the methodology, I am very unsure if your setting is really measuring what you intend. After reading your introduction, I supposed you would work with a company and compare different units against each other. However, you use students. The literature that raises serious doubts on tests on students is ample and I do agree that while this group is convenient, it comes with a lot of difficulties. As these members will all be comparable in age and work experience, a test of their psychological set-up is even more needed. 

Especially belonging to a team, trust in a team members etc. cannot be simulated by wearing a similar t-shirt. I understand that this is a repetitive game? Maybe men were more willing to try out a different strategy - irrespective of fear or trust?

As in general, psychological terms like fear and trust are not part of rational choice, and the prisoners dilemma is all about rational choice. These assumptions require much deeper rooting in literature and much more advanced explanations. 

The study would benefit greatly if you would introduce more control variables. Maybe differences in the majors of the students would explain differences, or general risk aversion, cultural programming etc. 

Your findings seem to proof what is already known. Eg. on line 605: for women, reducing within-group inequities is key, whereas for men maximizinbg group outcomes is more important. You then refer to 2 papers: one is a summary of other papers, the second shows that there are different distributions along the selfish - altruism spectrum between genders. A real proof that men are more interested in outcomes is not clear. 

Finally, I am unsure why you have chosen this journal over e.g. a gender-studies journal, for which the paper would be a better fit. 

Good luck!

Reviewer 3 Report

This is a very interesting study about the moderating role of gender in the relationship between team identity and fear response as well we sustainable cooperative behavior.

I have only some minor suggestions for you to improve your paper.

In the theory and hypothesis development section, you should clearly present what your dependent variables of interest are and how they are related to related past literature. As it currently stands, literature review and hypothesis development are mixed in an unorganized way, which can confuse readers.

Similar problems arise as readers follow through authors' logic about the moderating role of gender in the relationship between team identity and fear response as well we sustainable cooperative behavior.

This is a presentation issue. Accordingly I believe that authors can tackle these issues quite easily.

Good luck with your future work.

Best regards.

Round 2

Reviewer 1 Report

The reviewer appreciates the authors’ efforts to improve the paper, which has been significantly improved. However, some improvements still need to be made before the paper can be accepted.

Line 255 - "Experimental sessions were conducted in a laboratory and lasted approximately 50 minutes."  When did these sessions take place?

Line 624 - Review the sentence "We found the effect suggested by Simpson [9], that is, team identity does not affect free-riding response to fear , in women, but not in men."

Line 649 - "free riding" (?)

Least but not least, similarities were found with previous works. I think that in order to give greater emphasis to the novelty and originality of this paper, the authors' previous works should be mentioned and the differences should be highlighted.

Reviewer 2 Report

Dear authors,

thank you for sharing your paper with me. I acknowlege that you have improved your paper. Still, I maintain a critical stance towards you chosen methodology. This however is a topic that should we discussed with a wider community. Hence, I suggest to publish your text. 

all the best!

Author Response

Thank you very much.